# Synthesis, Antimicrobial, Anticancer, PASS, Molecular Docking, Molecular Dynamic Simulations & Pharmacokinetic Predictions of Some Methyl β-D-Galactopyranoside Analogs

**DOI:** 10.3390/molecules26227016

**Published:** 2021-11-20

**Authors:** Md. Ruhul Amin, Farhana Yasmin, Mohammed Anowar Hosen, Sujan Dey, Shafi Mahmud, Md. Abu Saleh, Talha Bin Emran, Imtiaj Hasan, Yuki Fujii, Masao Yamada, Yasuhiro Ozeki, Sarkar Mohammad Abe Kawsar

**Affiliations:** 1Laboratory of Carbohydrate and Nucleoside Chemistry, Department of Chemistry, Faculty of Science, University of Chittagong, Chittagong 4331, Bangladesh; ruhul@gmx.com (M.R.A.); yasminmunni39@gmail.com (F.Y.); m.a.hossain1996@gmail.com (M.A.H.); 2Department of Microbiology, Faculty of Biological Science, University of Chittagong, Chittagong 4331, Bangladesh; sujan.mbio@cu.ac.bd; 3Microbiology Laboratory, Department of Genetic Engineering and Biotechnology, University of Rajshahi, Rajshahi 6205, Bangladesh; shafimahmudfz@gmail.com (S.M.); saleh@ru.ac.bd (M.A.S.); 4Department of Pharmacy, BGC Trust University Bangladesh, Chittagong 4381, Bangladesh; talhabmb@bgctub.ac.bd; 5Department of Biochemistry and Molecular Biology, Faculty of Science, University of Rajshahi, Rajshahi 6205, Bangladesh; hasanimtiaj@yahoo.co.uk; 6School of Pharmaceutical Sciences, Nagasaki International University, 2825-7, Huis Ten Bosch-cho, Sasebo, Nagasaki 859-3298, Japan; yfujii@niu.ac.jp; 7School of Sciences, Yokohama City University, 22-2, Seto, Kanazawa-ku, Yokohama 236-0027, Japan; yamada.mas.ug@yokohama-cu.ac.jp

**Keywords:** methyl β-D-galactopyranoside, synthesis, PASS, molecular docking, molecular dynamics, Pharmacokinetic

## Abstract

A series of methyl β-D-galactopyranoside (MGP, **1**) analogs were selectively acylated with cinnamoyl chloride in anhydrous *N*,*N*-dimethylformamide/triethylamine to yield 6-*O*-substitution products, which was subsequently converted into 2,3,4-tri-*O*-acyl analogs with different acyl halides. Analysis of the physicochemical, elemental, and spectroscopic data of these analogs revealed their chemical structures. In vitro antimicrobial testing against five bacteria and two fungi and the prediction of activity spectra for substances (PASS) showed promising antifungal functionality comparing to their antibacterial activities. Minimum inhibition concentration (MIC) and minimum bactericidal concentration (MBC) tests were conducted for four compounds (**4**, **5**, **6**, and **9**) based on their activity. MTT assay showed low antiproliferative activity of compound **9** against Ehrlich’s ascites carcinoma (EAC) cells with an IC_50_ value of 2961.06 µg/mL. Density functional theory (DFT) was used to calculate the thermodynamic and physicochemical properties whereas molecular docking identified potential inhibitors of the SARS-CoV-2 main protease (6Y84). A 150-ns molecular dynamics simulation study revealed the stable conformation and binding patterns in a stimulating environment. In-silico ADMET study suggested all the designed molecules to be non-carcinogenic, with low aquatic and non-aquatic toxicity. In summary, all these antimicrobial, anticancer and in silico studies revealed that newly synthesized MGP analogs possess promising antiviral activity, to serve as a therapeutic target for COVID-19.

## 1. Introduction

Carbohydrates are important molecules in nature that play a variety of roles in biological processes. Scientists worldwide are searching for more effective and safe antimicrobial and antiviral agents to treat diseases caused by pathogenic organisms. For these reasons, the best way to develop effective antimicrobial and antiviral agents is to synthesize new chemicals and test their antimicrobial activity. For a long time, carbohydrates have been a very attractive topic for scientists due to their immense importance in biological systems, including viral and bacterial infections, cell growth and proliferation, cell-cell communication, and immune response [1,2]. They are the source of the metabolic energy supply and the fine-tuning of cell-cell interactions and other crucial processes [3,4]. It was found from the literature survey that a large number of biologically active compounds also possess aromatic, heteroaromatic, and acyl substituents [5,6,7,8,9,10,11,12]. For example, the benzene substituted benzene, and nitrogen, sulfur, and halogen-containing substituents are known to enhance the biological activity of the parent compound [13,14,15,16]. If an active nucleus is linked to another active nucleus, the resulting molecule may possess greater potential for biological activity [17,18]. Furthermore, selective acylation of carbohydrates and microbial activity evaluation [19,20] show that combining two or more heteroaromatic nuclei and acyl groups increases biological activity significantly more than its parent nucleus [21]. The recent outbreak of a novel coronavirus disease (COVID-19), caused by a severe acute respiratory syndrome (SARS)-like coronavirus, began in Wuhan, China, and is now considered a global pandemic [22]. Some monosaccharide analogs have been identified as potential inhibitors of cancer cell protein [23]. Main protease (Mpro) is a crucial non-structural protein that contributes to viral replication and maturation [24]. In a recent study, Liu et al. have ensured that the Mpro enzyme is present in the SARS-CoV-2 [25]. The genome of SARS-CoV-2 encodes pp1a and pp1b of two large polyproteins like another coronaviridae genome. Modifications of hydroxyl (-OH) group of nucleoside and monosaccharide structure revealed some potent SARS-CoV-2 candidates [26,27,28] and antimicrobial agents [29,30]. In this research, we have modified the hydroxyl (-OH) group of MGP by some acyl substituents (including aromatic and heteroaromatic groups). These modified analogs are optimized based on quantum mechanical methods to realize their thermal, electrical stability, and biochemical behavior. The free energy, enthalpy, dipole moment, HOMO-LUMO gap, chemical potentials, DOS plot, and molecular electrostatic potential were calculated to compare their thermal and chemical properties. All compounds were subjected to antimicrobial screening to predict their PASS properties. Molecular docking was performed against a receptor protein of SARS-CoV-2 main protease (PDB: 6Y84) to identify the binding mode, affinity, and non-bonding interaction of MGP analogs with the receptor protein. To confirm the stability of the docked complexes, molecular dynamics was performed for 150 ns. In addition, pharmacokinetic prediction has been performed to compare their absorption, metabolism, and toxicity. This study aimed to investigate the inhibitory effect of several synthesized MGP analogs against novel coronavirus through synthetic, spectral, anticancer and several in silico and computational approaches, namely DFT analysis, molecular docking, molecular dynamics, and targeting SARS-CoV-2 main protease as a prospective receptor.

## 2. Results

The present study performed regioselective cinnamoylation of methyl β-D-galactopyranoside (**1**) using the direct acylation method. The resulting cinnamoylation product was transformed into several derivatives employing various acylating agents. These MGP analogs were subjected to antimicrobial testing and geometrical optimization to determine the mode of their antimicrobial behavior. After screening modified MGP analogs for potential antimicrobial activity, the researchers discovered that most of the compounds have potent bactericidal and fungicidal in vitro activity against tested bacterial and fungal pathogens. The PASS web tool was also used to predict the biological activities of partial acylated derivatives. The observed activities were then rationalized using thermodynamic, MEP, molecular docking, molecular dynamics simulation, and combined in silico pharmacokinetics and drug-likeness properties calculations.

### 2.1. Characterization

Our initial effort was to conduct selective cinnamoylation of methyl β-D-galactopyranoside (**1**) with a unimolecular amount of cinnamoyl chloride in dry *N*,*N*-dimethylformamide, and triethylamine at −5 °C. The conventional work-up procedure, followed by removal of solvent and silica gel column chromatographic purification, we obtained the cinnamoyl derivative (**2**) in 89.85% yield as needles which was used in the next stage (Figure 1). 

This compound (**2**) was sufficiently pure for the next reactions to synthesize analogs **3**–**10** (Appendix A). The structure of the cinnamoyl derivative (**2**) was established by analyzing its elemental data, FTIR, and ^1^H-NMR spectra. The FTIR of this compound showed the following absorption bands: 1705 (C=O), 1628 (-CH=CH-), and 3413~3481 cm^−1^ (br, –OH) stretching. In its ^1^H-NMR spectrum, two one-proton doublet at δ 7.71 (as d, *J* = 12.0 Hz, PhCH=CHCO-) and δ 6.42(as d, *J* = 12.1 Hz, PhCH=C*H*CO-) due to the presence of one cinnamoyl group in the molecule. In addition, a two-proton multiplet at δ 7.46 (as m, Ar-H) and a three-proton multiplet at δ 7.32 (as, m, Ar-H) due to the one aromatic ring protons. The downfield shift of C-6 to δ 4.46 (as dd, *J* = 11.1 and 6.5 Hz, 6a) and 4.44 (as dd, *J* = 11.1 and 6.7 Hz, 6b) from its usual value (~4.00 ppm) [31] indicated the attachment of the cinnamoyl group at position 6.

The ^13^C-NMR spectrum also showed the presence of one cinnamoyl group by displaying the following expected resonance peaks: δ 164.81 (C_6_H_5_CH=CH*C*O-), 151.52 (C_6_H_5_*C*H=CH*C*O-), 131.90, 131.06, 128.70 (×2), 128.60 (×2) (*C*_6_H_5_CH=CHCO-), 122.11 (C_6_H_5_CH=*C*H*C*O-). Mass spectrum of compound (**2**) had a molecular ion peak at *m/z* [M+1]^+^ 325.33 correspondings to molecular formula, C_16_H_20_O_7_. Formation of compound (**2**) can be explained by attachment of cinnamoyl group to the more reactive and less sterically hindered primary -OH group at the C-6 position, with consequent formation of methyl 6-*O*-cinnamoyl-β-D-galactopyranoside (**2**) as sole product.

Further support for the structure of compound **2** was achieved by the preparation and identification of its butyryl derivative **3**. The ^1^H-NMR spectrum displayed two six-proton multiplets at δ 2.36 {3 × CH_3_CH_2_C*H*_2_CO-} and δ 1.55 {3 × CH_3_C*H*_2_CH_2_CO-} and nine-proton multiplet at δ 0.90 {3 × C*H*_3_(CH_2_)_2_CO-} due to three butyryl groups in the molecule. Also, C-2 proton deshielded to δ 5.21 (as dd, *J* = 8.2 and 10.5 Hz), C-3 proton to δ 5.08 (as dd, *J* = 3.1 and 10.5 Hz) and C-4 proton to δ 4.36 (as d, *J* = 3.5 Hz) as compared to the precursor triol (**2**) (δ 3.88) indicating the attachment of the butyryl groups at 2, 3 and 4 positions. Its ^13^C-NMR spectrum also showed the presence of three butyryl groups by displaying the following characteristic peaks: δ 174.55, 173.66, 173.01 3 × CH_3_(CH_2_)_2_*C*O-}, 34.18, 34.07, 32.11, 24.52, 24.44, 22.23 {3×CH_3_(*C*H_2_)_2_CO*-*}, 13.72, 13.80, 13.44 {3 × *C*H_3_(CH_2_)_2_CO*-*}. Molecular ion peak at *m/z* [M+1]^+^ 535.60 correspondings to molecular formula, C_28_H_38_O_10,_ and the structure of the tributyryate was ascertained as methyl 2,3,4-tri-*O*-butyryl-6-*O*-cinnamoyl-β-D-galactopyranoside (**3**). 6-*O*-Cinnamoyl derivative **2** was also similarly reacted with several fatty acid derivatives such as hexanoyl chloride, lauroyl chloride, myristoyl chloride, and palmitoyl chloride in dry DMF/Et_3_N to afford 2,3,4-tri-*O*-hexanoyl-β-D-galactopyranoside (**4**), 2,3,4-tri-*O*-lauroyl-β-D-galactopyranoside (**5**), 2,3,4-tri-*O*-myristoyl-β-D-galactopyranoside (**6**), and 2,3,4-tri-*O*-palmitoyl-β-D-galactopyranoside (**7**) with good yields, respectively. The structures of these derivatives were ascertained by complete interpretation of their IR, ^1^H-NMR, ^13^C-NMR, mass, elemental analysis. Final confirmation of the structure of the cinnamoyl derivative **2** was carried out by conversion of its trityl derivative (**8**), 4-nitrobenzoyl derivative (**9**), and 4-bromobenzoyl derivative (**10**). FTIR spectrum of compound **8** showed the following characteristic absorption bands: 1684 cm^−1^ (-CO stretching), 1632 cm^−1^ (-CH=CH-stretching) and its ^1^H-NMR spectrum, the presence of eighteen-proton multiplet at δ 7.53 (3 × Ar-H)) and a twenty seven-proton multiplet at δ 7.31 (3 × Ar-H) were indicative of the introduction of three trityl groups in the molecule. In the FTIR spectrum of compound **9**, the bands at 1711, 1635 cm^−1^ corresponded to the carbonyl and (-CH=CH-) stretchings. Complete analysis of the ^1^H-NMR spectrum enabled us to assign the structure of the 4-nitrobenzoyl derivative as methyl 6-*O*-cinnamoyl-2,3,4-tri-*O*-(4-nitrobenzoyl)-β-D-galactopyranoside (**9**). The ^1^H-NMR spectrum of compound **10** showed the two six-aromatic proton multiplets at δ 8.01 (as m) and δ 7.93 (as m), which are characteristic of the *p*-substituted benzoyl group. Furthermore, the deshielding of C-2, C-3, and C-4 protons from their usual values (compound **2**) and the resonance of other protons in their anticipated positions, ^13^C-NMR, and mass spectra confirmed the structure of this compound as methyl 2,3,4-tri-*O*-(4-bromobenzoyl)-6-*O*-cinnamoyl-β-D-galactopyranoside (**10**). Furthermore, these acylation products and their derivatives were found to have a long shelf life. Therefore, these products may further be utilized as probable starting materials for the synthesis of newer derivatives.

### 2.2. In Vitro Antibacterial Activities

The results of the disc diffusion test revealed that most of the synthesized compounds were active against the tested Gram-positive bacteria (Table 1). The synthesized compound **9** exhibited the highest zone of inhibition against *Bacillus subtilis* (22 ± 0.3 mm) extensively, and *Staphylococcus aureus* (30 ± 0.4 mm) and compound **10** showed notable zone of inhibition against *Bacillus subtilis* (20 ± 0.3 mm) and *Staphylococcus aureus* (23 ± 0.4 mm). These compounds **9** and **10** remarkably showed a higher zone of inhibition than the standard antibiotic azithromycin against *Bacillus subtilis* (19 ± 0.3 mm) and *Staphylococcus aureus* (18 ± 0.3 mm). The other compounds (**2**–**8**) exhibited fluctuation in inhibition against the Gram-positive bacteria *Bacillus subtilis* and *Staphylococcus aureus*. Based on Gram-positive antibacterial activity, compounds can be ordered as 9 ˃ 10 ˃ 5 ˃ 4 ˃ 6 ˃ 3 ˃ 2 = 8 ˃ 7 (Appendix A). The effects of the synthesized compounds’ disc diffusion test (Table 2) against the tested three Gram-negative bacteria were also very excellent. Where compound **5** exhibited the highest zone of inhibition against *Escherichia coli* (21 ± 0.3 mm), *Salmonella abony* (27 ± 0.4 mm), and *Pseudomonas aeruginosa* (27 ± 0.4 mm) extensively. For compound **5**, the zone of inhibition was largely higher than the standard antibiotic azithromycin against *Escherichia coli* (17 ± 0.3 mm), *Salmonella abony* (19 ± 0.3 mm), and *Pseudomonas aeruginosa* (17 ± 0.3 mm). 

The set of compounds **3**, **4**, **6**, and **9** exhibited a zone of inhibition against all the three Gram-negative bacteria *Escherichia coli, Salmonella abony*, and *Pseudomonas aeruginosa*, and compound **10** showed inhibition against only two bacteria, *Escherichia coli*, and *Salmonella abony*. Based on activity compounds can be ordered as 5 ˃ 3 ˃ 4 ˃ 6 ˃ 9 ˃ 10 ˃ 3 > 2 = 8 ˃ 7 these results are presented pictorially in Appendix A and graphically in Appendix A.

### 2.3. Determination of MIC and MBC 

The most active compounds **4**, **5**, **6**, and **9** against most bacterial pathogen are scanned for further investigation of minimum inhibitory concentration (MIC) and minimum bactericidal concentration (MBC). As shown in Figure 1, the lowest value of MIC was found for compound **5** (0.352 ± 0.01mg), inhibiting *S. aureus, S. abony,* and *P. aeruginosa*. Whereas the highest MIC values (1.172 ± 0.03 mg) was obtained for compound **4** and **9** against *E. coli* and compound **4** also showed the same MIC for *P. aeruginosa*. The MIC value (1.406 ± 0.02 mg) for compounds **4** and **5** against *B. subtilis* and compound **6** against *S. aureus, S. abony,* and *P. aeruginosa* lie in the ranges (0.352 ± 0.01–1.172 ± 0.03 mg). After determining the minimum inhibitory concentration (MIC) for compounds **4**, **5**, **6**, and **9** against the bacterial pathogen, the minimum bactericidal concentration (MBC) was also determined where 99% of pathogens were killed. The lowest value of MBC was found for compound **5** (704 ± 0.02 mg) destructing *S. aureus* and *P. aeruginosa*. Whereas the highest MBC values (2.812 ± 0.06 mg) were obtained for compound **4** against *E. coli*, *B. subtilis*, and compound **9** showed the same MBC for killing *B. subtilis* and *P. aeruginosa*. This maximum MBC value was also found for compound **5** (2.812 ± 0.06 mg) against the *B. subtilis* and compound **6** killing *S. aureus, S. abony*, and *P. aeruginosa*. The MBC values for these compounds killing the other tested organism lie between the ranges (0.704 ± 0.02–2.812 ± 0.06 mg). MBCs are also presented graphically in Figure 2. 

### 2.4. Antifungal Evaluation

Most of the methyl β-D-galactopyranoside (**1**) derivatives had been found to have outstanding inhibition to the mycelial growth of both *A. niger* and *A. flavus* (Table 3). Among the tested derivatives, **10** inhibited 91.67 ± 1.2.2% against the *A. niger* and 82.77 ± 1.2 % against the *A. flavus* in their potential antifungal assessment. Remarkable mycelial growth prevention was also built up for compound **4** against the *A. niger* (86.67 ± 1.2) and *A. flavus* (79.44 ± 1.1%) in their mycelial growth test. It was observed that compounds **3**, **4**, **5**, **6**, and **10** were very much effective against the *A. niger* and *A. flavus,* and their zone of inhibition is higher than standard antibiotic nystatin (Appendix A). The observed results revealed that the presence of different acyl moieties, including the 3-chlorobenzoyl, butyryl, octanoyl, lauroyl, trityl, and 3-bromobenzoyl groups, significantly enhanced the antimicrobial activity of carbohydrates species.

### 2.5. Anticancer Screening 

MTT assay was used to investigate the effect of compound **9** (4-nitrobenzoyl derivative) in vitro on EAC cells.

The EAC cell death took place in a dose-dependent manner (as shown in Figure 3). At 500 µg/mL protein concentration, the inhibitory effect of compound (**9**) was 8.77%, whereas at 250, 125 and 62.5 µg/mL were 3.29%, 2.08% and 1.02%, respectively. When the concentrations were gradually reduced, the compound’s inhibitory effect decreased, and the IC_50_ value was determined to be 2961.06 µg/mL.

### 2.6. Assessment of Antimicrobial Activities: PASS

We have also predicted the antimicrobial spectrum applying web server PASS of all the MGP analogs **2**–**10**. The PASS results are yclept as Pa and Pi, which are displayed in Appendix A. As can be seen in predication Appendix A, MGP analogs **2**–**10** showed 0.43 < Pa < 0.55 for antibacterial, 0.61 < Pa < 0.71 for antifungal, 0.47 < Pa < 0.64 for antioxidant and 0.57 < Pa < 0.83 for anti-carcinogenic properties. These results revealed that these molecules were more efficient against fungi in comparison with that of bacterial pathogens. Attachment of additional aliphatic acyl chains (C4 to C16) increased antifungal activity (Pa = 0.713) of MGP (**1**, Pa = 0.628), whereas insertion of Br- and NO_2_- substituted aromatic groups also improve reasonably. The same scenario was observed for an antioxidant activity where acyl chain analogs revealed improves values than the halo-benzoyl analogs. However, analog **2**, which has the cinnamoyl group, exhibited the highest antioxidant activity (Pa = 0.647). We also tried to predict the anti-carcinogenic parameter of these analogs. Therefore PASS determination exhibited 0.57 < Pa < 0.83 for anti-carcinogenic, which revealed that the MGP analogs were more potential as anti-carcinogenic agents than previous antimicrobial parameters. Significantly, antibacterial, antifungal, antioxidant, and anti-carcinogenic properties of MGP analogs with saturated acyl chains (**3**–**7**) were found more promising than the halo-benzoyl analog (**2** and **8**–**10**).

### 2.7. Thermodynamic Analysis

The free energy and enthalpy values can be used to calculate the spontaneity of a reaction and the stability of a product [32]. The dipole moment influences hydrogen bond formation and nonbonded interactions in drug design [33]. Free energy (*G*) is an important criterion for representing the interaction of binding partners, with a negative value indicating that spontaneous binding and interaction are preferable. In the present study, all the MGP analogs possess a greater negative value for *E*, *H*, and *G* than the parent MGP, and hence, indicated that the attachment of the ester group could improve the interaction and binding of these molecules with different microbial communities enzymes. The highest free energy is (−9702.8701 Hartree) observed for MGP analog (**10**), which also showed the highest enthalpy (−9702.7440 Hartree) and highest electronic energy (−9702.7449 Hartree) (Appendix A). 

The dipole moment of MGP analogs (**3**–**5**) and (**8**–**10**) was higher than the MGP, resulting from their better binding affinity and interactions with the amino acid residues of the receptor protein. The highest dipole moment is (6.4697 Debye) observed for analog (**10**), whereas (**3**) showed the lowest value (3.1229 Debye). These scores are gradually increased with longer carbon chains (**1**–**8**). Halogenated and aromatic analogs had better scores for all parameters, as evidenced by analogs (**9**) (4-NO_2_.Bz) and (**10**) (4-Br.Bz), which had the highest free energy of the therapeutics under investigation and showed markedly improved dipole moment. Finally, this discussion proves that modification of hydroxyl (-OH) groups of MGP significantly increases its thermodynamic properties, indicating the synthesized analogs’ inherent stability.

### 2.8. Frontier Molecular Orbitals (FMO) 

The most important molecular orbitals in a molecule are the frontier molecular orbitals, used to study chemical reactivity and kinetic stability. The highest occupied molecular orbital (HOMO) and the lowest unoccupied molecular orbital (LUMO) are the frontier molecular orbitals (LUMO). The electronic absorption relates to the transition from the ground to the first excited state and is mainly described by one electron excitation from HOMO to LUMO [34]. The HOMO and LUMO energies, HOMO-LUMO gap (∆), hardness (*η*), softness (*S*), and chemical potential (*µ*) index of all analogs are presented in Appendix A.

We observed that, with the increase of the number of ester groups and chain length (**2**–**10**) hardness, these compounds gradually decreased while their softness gradually improved. All these properties may show higher chemical activity and polarizability in the drug-related chemical and biochemical functionalities. For example, in Figure 4, the LUMO plot of the analog (**7**) showed that the electron was localized at the modified acylating group regions only. In contrast, the HOMO plot showed that the electron was localized on the upper part of the pyranose ring. 

### 2.9. Molecular Electrostatic Potential (MEP) 

The molecular electrostatic potential (MEP) is globally used as a reactivity map displaying the most suitable region for the electrophilic and nucleophilic attack of charged point-like reagents on organic molecules [35]. It helps to interpret the biological recognition process and hydrogen bonding interaction [36]. The significance of MEP lies in the fact that it simultaneously displays a molecular size, shape, as well as positive, negative, and neutral electrostatic potential regions in terms of color grading and is very useful in the research on molecular structure and physicochemical properties relationships [37]. 

The different values of electrostatic potential represent by different colors (Appendix A). Potential increases in the order red < orange < yellow < green < blue. Red color displays maximum negative area, which shows a favorable site for an electrophilic attack. The blue color indicates the maximum positive area favorable for nucleophilic attack. The green color represents zero potential areas.

### 2.10. Molecular Docking Simulation

The outcomes of the docking analysis showed that all analogs, along with the parent compound, obtain binding affinities ranging from −6.8 to −8.8 kcal/mol. As shown in Table 4, all MGP analogs (**2**–**7** and **8**–**10**) displayed higher binding affinity than parent MGP. These results indicated that modification of –OH group with a long carbon chain/aromatic ring molecule increased the binding affinity. 

Adding hetero groups like -Br and -NO_2_ made some fluctuations in binding affinities; however, modification with halogenated aromatic rings increased the binding affinity. The docked pose clearly showed that the drugs molecules bind within the SARS-CoV-2 M^pro^ macromolecular structure (Figure 5). 

The interactions between the inhibitor and bordering residues of SARS-CoV-2 M^pro^ are illustrated in the 2D schematics, which they were obtained by importing docking results into the Discovery Studio Visualizer (Figure 6), shows the amino acids participated in the pattern of interactions between the ligand and protein with an important contribution to the total energy of interaction. Most of these interactions include hydrophobic contacts, Vander Waals interactions, hydrogen bonds, electrostatic, carbonyl, and one specific atom-aromatic ring and provide insight into understanding molecular recognition (Table 5). Figure 5 and Appendix A depicts the docked conformation of the most active molecules (**3** and **10**) based on docking studies. As shown in Figure 6 and Appendix A, analogs (**3** and **4**) bind firmly through conventional hydrogen bonds with residues HIS41, GL143, besides other interactions such as carbon-hydrogen bonds (THR26, ASN142, and HIS41), alkyl and pi-alkyl (CYS145, HIS41, and MET165) interactions. Compound (**5**-**6**) showed similar binding sites with ARG298 (shorter distance 2.046 Å), CYS145, HIS41, GLY143, ASP153, PHE294, PRO293, and VAL104. MGP analogs (**5** and **6**) displayed more fluctuating binding scores than aliphatic analogs with new binding sites. Aromatic analogs (**7** and **9**–**10**) exhibited a higher binding score but a similar active site as aliphatic analogs.

Acyl chain substituted analogs (**3**–**6**) revealed a lower binding score with the main protease that indicates the burying of the ligand in the receptor cavity. Despite having lower binding affinity, they also interact with the catalytic binding of the main protease. So, these outcomes clear that, due to having high electron density, aromatic substituents can easily increase the binding ability and the antiviral ability of the MGP analogs. Along with PHE294, all the analogs displayed the maximum π- π interactions with the HIS41, MET49 denoting the tight binding with the active site. Reports suggest that PHE294 is considered the principal component of the PPS, PA, and PDH responsible for the accessibility of small molecules to the active site. Binding energy and binding mode were improved in that analogs (**2**–**7** and **8**–**10**) because of significant hydrogen bonding. Hydrogen bonds execute a vital function in shaping the specificity of ligand binding with the receptor, drug design in chemical and biological processes, molecular recognition, and biological activity [38]. 

It has already been reported that ten commercial medicines possibly form H-bonds with key residues of 2019-nCoV main protease [39]. Hydrogen bond surface and hydrophobic surface of analog (**10**) consequently represent in Appendix A. We observe that the analyzed MGP analogs bind within the active site of the main protease of SARS-CoV-2, like the standard drug Remdesivir, which is neccessary to prevent the protein mutarotation of the virus by minimizing the viral replication. Although the blind docking studies reveal that all the molecules can act as potential agents for COVID treatments, from the estimated free energy of binding values could infer that the analogs (**10**) with the highest negative minimum binding energy value −8.8 kcal/mol amongst all the studied analogs could be the best possible SARS-CoV-2 inhibitor.

### 2.11. Molecular Dynamics

The root means square deviation of the C-alpha atoms from the simulations systems was analyzed to understand the structural variations and the stability. 

Figure 7a indicates that the compounds **3**, **4**, **7**, **9**, **10** had the initial upper trend responsible for the flexible nature of the complexes at the initial phase. Therefore, the complexes reached the steady-state after 15 ns and maintained stability until the simulations’ last segment. Compound **4** had comparatively higher RMSD than other complexes, indicating the more flexible nature of these complexes than other compounds. As a result, all of the complexes had RMSD less than 2.5 during the simulation times, defining the complexes’ stable comparative nature. Furthermore, the solvent-accessible surface area of the simulation complexes was investigated to comprehend the changes in surface area of the complexes, where the higher SASA defines the extension of the surface volumes. In contrast, the lower SASA defines the truncated nature of the complexes. Figure 7b indicates that all complexes had relatively stable SASA profiles at the simulation trajectories. In addition, they experienced a lower degree of deviations, which indicated the stable nature of the complexes. Complexes 7 had lower SASA than other complexes, which defines these protein complexes’ condensed nature upon binding with compound **7**. Also, the radius of gyration of the complexes was analyzed, which defines the flexible and mobile nature of the complexes. Figure 7c indicates that the complexes had a steady trend of Rg except compound **7**. Also, the hydrogen bond of the simulation systems defines the stability of the complexes where all of the complexes had a stable trend. The root means square fluctuations or RMSF were also explored to understand the flexibility of amino acid residues. Figure 7d indicates that the maximum residues had a lower RMSF profile except for Gly3, Thr25, Arg61, Asn73, Arg223, and Thr305. The maximum residues had lower RMSF than 2.5Å, which defines the rigid state of the complexes. The hydrogen bond of a biological system requires assessment to evaluate the bonding and structural change in the complex.

They play a key role in giving the structural integrity of the systems. The five simulated complexes had a solid hydrogen-bonding pattern as lesser aberrations were observed. The number of hydrogen bonds between solute and solvents was calculated in Figure 7e. The simulation time’s initial and final phases also had the firm hydrogen bond as they did not fluctuate either.

### 2.12. Pharmacokinetic Profile, Toxicity, and Molecular Radar

Drug absorption depends on factors including membrane permeability [indicated by the cell line of colon cancer (Caco-2)], intestinal absorption, skin permeability thresholds, substrate or inhibitor of P-glycoprotein. The value of intestinal absorbance below 30% suggests poor absorbance. Table 6 shows that all of the analogs have excellent absorption with more than 30. Skin permeability is an important factor to consider when improving drug efficacy, and it is especially important in the development of transdermal drug delivery. A molecule will barely penetrate the skin if log Kp is more than −2.5 cm/h [40]. From Table 6, it can be seen that the skin permeability (Kp of MGP analogs is −2.735 cm/h (<−2.5). For the pkCSM predictive model, high Caco-2 permeability is translated into predicted log Papp values > 0.90 cm/s. As Table 6 shows, the value of Caco-2 permeability (log Papp) of the MGP analogs ranged from −5.1 to −2.4 cm/s, log Papp < 0.9 cm/s, so it is predicted that these have low Caco-2 permeability.

Highly water solubility was useful for delivering sufficient active ingredients in a small volume of such pharmaceutical dosage. These values water solubility is given in log (mol/L) (Insoluble ≤ −10 < poorly soluble < −6 < Moderately < −4 < soluble < −2 < very soluble < 0 ≤ highly soluble). From the results that appear in Table 6, it was observed that the analogs tested are soluble. Distribution volume (Vd) is a pharmacokinetic parameter reflecting the tendency of an individual substance to either linger in the plasma or redistribute to another tissue compartment. According to Pires et al. [40] VDss is considered low if it is below 0.71 L/kg (log VDss < −0.15) and high if it is above 2.81 L/kg (log VDss > 0.45). It can be shown from Table 7 that the value of MGP analogs VDss ranged from −1.646 to 0.359, with only one analog having a VDss value of < −0.15 (number 10). Blood-brain partitioning and brain distribution are critical properties for drugs targeting the central nervous system. 

The compounds tested a logBB < −1 considered poorly distributed to the brain. From Table 7 it can be seen that the logPS (the central nervous system (CNS) permeability) value of MGP analog range from −1.70 to −3.67, logPS < −3, so it can be predicted that analogs (**3**−**7**) are unable to penetrate the CNS. Furthermore, it can be seen from Table 7 the log CLtot value of MGP analogs ranges from −2.7 to 2.0 mL/min/kg, and from those values can be predicted the rate of excretion of the compound.

From Appendix A it can be seen that all MGP analogs do not affect or inhibit all the enzymes except CYP3A4, so it can be predicted that all of the analogs in the body tend to be metabolized by the P450 enzyme. Radar for bioactivity Charts of MGP analogs (Appendix A) revealed that all candidates had promising pharmacokinetic profiles. The pkCSM pharmacokinetics model predicts a given compound’s total clearance log(CLtot) in log(mL/min/kg). The higher the compound’s CLtot value, the faster the excretion processes. The compounds’ results are described in Appendix A, and their high LD50 values (2.33 to 2.65) indicate that the compounds are lethal only at extremely high doses. A negative AMES test result indicates that the compound is not mutagenic. The findings also suggest that none of the analogs tested inhibited the hERG channel and did not cause skin sensitization. 

## 3. Materials and Methods

### 3.1. General Information

Melting temperatures were determined on an electrothermal melting point apparatus (England) and are uncorrected. Thin-layer chromatography (TLC) was performed on Kieselgel GF_254,_ and spots were detected by spraying with 1% H_2_SO_4_, followed by heating at 150–200 °C. Column chromatography was performed with silica gel G_60_. ^1^H-NMR (400 MHz) and ^13^C-NMR (100 MHz) spectra were recorded for solutions in CDCl_3_ with TMS as an internal standard at the WMSRC Jahangirnagar University, Savar, Dhaka, Bangladesh. Infrared spectral analyses were recorded using a Fourier-transform infrared (FTIR) spectrophotometer (IR Prestige-21, Shimadzu, Japan) within 200–4000 cm^−1^ at the Department of Chemistry, University of Chittagong, Bangladesh. Mass spectra of the synthesized compounds were obtained by liquid chromatography-electrospray ionization tandem mass spectrometry in positive ionization mode. All evaporations were conducted under reduced pressure using Büchi rotary evaporator (Germany). 

### 3.2. Synthesis

A cooled (0 °C) and stirred solution of the methyl β-D-galactopyranoside (**1**) (100 mg, 0.515 mmol), in anhydrous *N*, *N*-dimethylformamide (3 mL), and triethylamine (0.15 mL) was treated with cinnamoyl chloride (185 mg, 1.1 molar eq.). Stirring was continued for 6 h at 0 °C and overnight at room temperature. The progress of the reaction was monitored by TLC (CH_3_OH-CHCl_3_, 1:6), which indicated full conversion of the starting material into a single product (*R_f_* = 0.52). The resulting syrup was passed through a silica gel column and eluted with CH_3_OH-CHCl_3_ (1:6) provided the cinnamoyl analog (**2**) (150 mg, 89.85%) as crystalline solid. Recrystallization from ethyl acetate-hexane gave the methyl 6-*O*-cinnamoyl-β-D-galactopyranoside (**2**) as needless, m.p. 73–75 °C. Thus, the compound was sufficiently pure for use in the next stage without further purification and identification.

Methyl 6-*O*-cinnamoyl-β-D-galactopyranoside (2): IR (KBr): ν/cm^−1^ 1705 (C=O), 1628 (-CH=CH-), 3413~3481 cm^−1^ (-OH); ^1^H-NMR (400 MHz, CDCl_3_) (ppm): δ_H_ 7.71 (1H, d, *J* = 12.0 Hz, PhC*H*=CHCO-), 7.46 (2H, m, Ar-H), 7.32 (3H, m, Ar-H), 6.42 (1H, d, *J* = 12.1 Hz, PhCH=C*H*CO-), 5.10 (1H, d, *J* = 8.0 Hz, H-1), 4.46 (1H, dd, *J* = 11.1 and 6.5 Hz, H-6a), 4.44 (1H, dd, *J* = 11.1 and 6.7 Hz, H-6b), 4.18 (1H, d, *J* = 3.5 Hz, H-4), 4.00 (1H, dd, *J* = 3.0 and 10.5 Hz, H-3), 3.88 (1H, dd, *J* = 8.0 and 10.5 Hz, H-2), 3.77 (1H, m, H-5), 3.46 (3H, s, 1-OC*H*_3_); ^13^C-NMR (100 MHz, CDCl_3_): δ_C_ 164.81 (C_6_H_5_CH=CH*C*O-), 151.52 (C_6_H_5_*C*H=CH*C*O-), 131.90, 131.06, 128.70 (×2), 128.60 (×2) (*C*_6_H_5_CH=CHCO-), 122.11 (C_6_H_5_CH=*C*H*C*O-), 97.07 (C-1), 72.91 (C-2), 71.33 (C-4), 70.66 (C-3), 69.39 (C-5), 63.01 (C-6), 54.26 (1-O*C*H_3_). LC-MS [M+1]^+^ 325.33. Anal Calcd. for C_16_H_20_O_7_: C, 59.24, H, 6.21%; found: C, 59.25, H, 6.23%. 

### 3.3. General Procedure of the Synthesis of 6-O-cinnamoyl Analogs

A solution of the cinnamoyl analog (**2**) (150 mg, 0.46 mmol) in dry *N*,*N*-dimethylformamide (3 mL) and triethylamine (0.15 mL) was cooled to 0 °C when butyryl chloride (0.23 mL, 5.0 molar eq.) was added. The mixture was stirred at 0 °C for 6 h and then overnight at room temperature. TLC (CH3OH-CHCl3, 1:5) indicated the complete conversion of the starting material into the faster-moving product (*R_f_* = 0.53). The conventional work-up procedure followed by silica gel column chromatography purification with CH_3_OH-CHCl_3_ (1:5 as eluent) afforded the butyryl analog (**3**) (196 mg, 79.27%) as a crystalline solid, m.p. 105–107 °C (EtOAc-C_6_H_14_). Similar reaction and purification procedure was applied to prepare compound **4** (94.94%) as needles, m.p. 125–126 °C (EtOAc-C_6_H_14_), compound **5** (84.19%) as needles, m.p. 111–113 °C (EtOAc-C_6_H_14_), compound **6** (86.0%) as crystalline solid, m.p. 135-136^0^C (EtOAc-C_6_H_14_), compound **7** (84.29%) as needles, m.p. 151–154 °C (EtOAc-C_6_H_14_), compound **8** (92.94%) as crystalline solid, m.p. 138–140 °C (EtOAc-C_6_H_14_), compound **9** (97.68%) as needles, m.p. 144–145 °C (EtOAc-C_6_H_14_) and compound **10** (95.29%) as needles, m.p. 128–130 °C (EtOAc-C_6_H_14_). 

Methyl 2,3,4-tri-*O*-butyryl-6-*O*-cinnamoyl-β-D-galactopyranoside (3): IR (KBr): ν/cm^−1^ 1702 (C=O), 1626 (-CH=CH-); ^1^H-NMR (400 MHz, CDCl_3_) (ppm): δ_H_ 7.61 (1H, d, *J* = 12.1 Hz, PhC*H*=CHCO-), 7.43 (2H, m, Ar-H), 7.29 (3H, m, Ar-H), 6.41 (1H, d, *J* = 12.1 Hz, PhCH=C*H*CO-), 5.56 (1H, d, *J* = 8.1 Hz, H-1), 5.21 (1H, dd, *J* = 8.2 and 10.5 Hz, H-2), 5.08 (1H, dd, *J* = 3.1 and 10.5 Hz, H-3), 4.36 (1H, d, *J* = 3.5 Hz, H-4), 4.14 (1H, dd, *J* = 11.2 and 6.7 Hz, H-6a), 4.01 (1H, dd, *J* = 11.1 and 6.8 Hz, H-6b), 3.56 (1H, m, H-5), 3.17 (3H, s, 1-OC*H*_3_), 2.36 {6H, m, 3×CH_3_CH_2_C*H*_2_CO-}, 1.55 (6H, m, 3×CH_3_C*H*_2_CH_2_CO-), 0.90 {9H, m, 3×C*H*_3_(CH_2_)_2_CO-}; LC-MS [M+1]^+^ 535.60; Anal Calcd. for C_28_H_38_O_10_: C, 62.90, H, 7.16%; found: C, 62.89, H, 7.18%.

Methyl 6-*O*-cinnamoyl-2,3,4-tri-*O*-hexanoyl-β-D-galactopyranoside (4): IR (KBr): ν/cm^−1^ 1711 (C=O), 1621 (-CH=CH-); ^1^H-NMR (400 MHz, CDCl_3_) (ppm): δ_H_ 7.76 (1H, d, *J* = 12.1 Hz, PhC*H*=CHCO-), 7.51 (2H, m, Ar-H), 7.33 (3H, m, Ar-H), 6.44 (1H, d, *J* = 12.1 Hz, PhCH=C*H*CO-), 5.04 (1H, d, *J* = 8.1 Hz, H-1), 4.88 (1H, dd, *J* = 8.1 and 10.6 Hz, H-2), 4.74 (1H, dd, *J* = 3.1 and 10.6 Hz, H-3), 4.65 (1H, d, *J* = 3.6 Hz, H-4), 4.12 (1H, dd, *J* = 11.2 and 6.6 Hz, H-6a), 3.88 (1H, dd, *J* = 11.2 and 6.8 Hz, H-6b), 3.58 (1H, m, H-5), 3.21 (3H, s, 1-OC*H*_3_), 2.33 {6H, m, 3×CH_3_(CH_2_)_3_C*H*_2_CO-}, 1.62 {6H, m, 3×CH_3_(CH_2_)_2_C*H*_2_CH_2_CO-}, 1.28 {12H, m, 3×CH_3_(C*H*_2_)_2_CH_2_CH_2_CO-}, 0.88 {9H, m, 3×C*H*_3_(CH_2_)_4_CO-}; LC-MS [M+1]^+^ 619.75; Anal Calcd. for C_34_H_50_O_10_: C, 65.99, H, 8.14%; found: C, 65.97, H, 8.13%.

Methyl 6-*O*-cinnamoyl-2,3,4-tri-*O*-lauroyl-β-D-galactopyranoside (5): IR (KBr): ν/cm^−1^ 1710 (C=O), 1621 (-CH=CH-); ^1^H-NMR (400 MHz, CDCl_3_) (ppm): δ_H_ 7.72 (1H, d, *J* = 12.1 Hz, PhC*H*=CHCO-), 7.50 (2H, m, Ar-H), 7.41 (3H, m, Ar-H), 6.46 (1H, d, *J* = 12.2 Hz, PhCH=C*H*CO-), 5.41 (1H, d, *J* = 8.1 Hz, H-1), 5.21 (1H, dd, *J* = 8.1 and 10.6 Hz, H-2), 5.11 (1H, dd, *J* = 3.1 and 10.5 Hz, H-3), 4.30 (1H, d, *J* = 3.5 Hz, H-4), 4.21 (1H, dd, *J* = 11.2 and 6.8 Hz, H-6a), 4.01 (1H, dd, *J* = 11.2 and 6.8 Hz, H-6b), 3.96 (1H, m, H-5), 3.41 (3H, s, 1-OC*H*_3_), 2.34 {6H, m, 3×CH_3_(CH_2_)_9_C*H*_2_CO-}, 1.62 {6H, m, 3×CH_3_(CH_2_)_8_C*H*_2_CH_2_CO-}, 1.28 {48H, m, 3×CH_3_(C*H*_2_)_8_CH_2_CH_2_CO-}, 0.89 {9H, m, 3×C*H*_3_(CH_2_)_10_CO-}; LC-MS [M+1]^+^ 875.25; Anal Calcd. for C_52_H_86_O_10_: C, 71.43, H, 9.91%; found: C, 71.44, H, 9.92%.

Methyl 6-*O*-cinnamoyl-2,3,4-tri-*O*-myristoyl-β-D-galactopyranoside (6): IR (KBr): ν/cm^−1^ 1715 (C=O), 1624 (-CH=CH-); ^1^H-NMR (400 MHz, CDCl_3_) (ppm): δ_H_ 7.72 (1H, d, *J* = 12.1 Hz, PhC*H*=CHCO-), 7.50 (2H, m, Ar-H), 7.41 (3H, m, Ar-H), 6.46 (1H, d, *J* = 12.2 Hz, PhCH=C*H*CO-), 5.45 (1H, d, *J* = 8.0 Hz, H-1), 5.26 (1H, dd, *J* = 8.2 and 10.5 Hz, H-2), 5.03 (1H, dd, *J* = 3.2 and 10.6 Hz, H-3), 4.38 (1H, d, *J* = 3.5 Hz, H-4), 4.21 (1H, dd, *J* = 11.1 and 6.8 Hz, H-6a), 4.06 (1H, dd, *J* = 11.1 and 6.8 Hz, H-6b), 3.94 (1H, m, H-5), 3.48 (3H, s, 1-OC*H*_3_), 2.28 {6H, m, 3×CH_3_(CH_2_)_11_C*H*_2_CO-}, 1.56 {6H, m, 3×CH_3_(CH_2_)_10_C*H*_2_CH_2_CO-}, 1.28 {60H, m, 3×CH_3_(C*H*_2_)_10_CH_2_CH_2_CO-}, 0.87 {9H, m, 3×C*H*_3_(CH_2_)_12_CO-}; LC-MS [M+1]^+^ 956.39; Anal Calcd. for C_58_H_98_O_10_: C,72.91, H, 10.33%; found: C, 72.93, H, 10.35%.

Methyl 6-*O*-cinnamoyl-2,3,4-tri-*O*-palmitoyl-β-D-galactopyranoside (7): IR (KBr): ν/cm^−1^ 1702 (C=O), 1628 (-CH=CH-); ^1^H-NMR (400 MHz, CDCl_3_) (ppm): δ_H_ 7.70 (1H, d, *J* = 12.1 Hz, PhC*H*=CHCO-), 7.52 (2H, m, Ar-H), 7.42 (3H, m, Ar-H), 6.42 (1H, d, *J* = 12.2 Hz, PhCH=C*H*CO-), 5.32 (1H, d, *J* = 8.0 Hz, H-1), 5.12 (1H, dd, *J* = 8.2 and 10.6 Hz, H-2), 4.92 (1H, dd, *J* = 3.0 and 10.6 Hz, H-3), 4.75 (1H, d, *J* = 3.6 Hz, H-4), 4.26 (1H, dd, *J* = 11.1 and 6.8 Hz, H-6a), 4.11 (1H, dd, *J* = 11.2 and 6.8 Hz, H-6b), 4.02 (1H, m, H-5), 3.28 (3H, s, 1-OC*H*_3_), 2.35 {6H, m, 3×CH_3_(CH_2_)_13_C*H*_2_CO-}, 1.26 {78H, m, 3×CH_3_(C*H*_2_)_13_CH_2_CO-}, 0.91 {9H, m, 3×C*H*_3_(CH_2_)_14_CO-}; LC-MS [M+1]^+^ 1040.55; Anal Calcd. for C_64_H_110_O_10_: C, 73.94, H, 10.66%; found: C, 73.95, H, 10.68%. 

Methyl 6-*O*-cinnamoyl-2,3,4-tri-*O*-trityl-β-D-galactopyranoside (8): IR (KBr): ν/cm^−1^ 1684 (C=O), 1632 (-CH=CH-); ^1^H-NMR (400 MHz, CDCl_3_) (ppm): δ_H_ 7.68 (1H, d, *J* = 12.1 Hz, PhC*H*=CHCO-), 7.53 (18H, m, 3×Ar-H), 7.47 (2H, m, Ar-H), 7.40 (3H, m, Ar-H), 7.31 (27H, m, 3×Ar-H), 6.40 (1H, d, *J* = 12.2 Hz, PhCH=C*H*CO-), 5.41 (1H, d, *J* = 8.2 Hz, H-1), 5.21 (1H, dd, *J* = 8.3 and 10.6 Hz, H-2), 5.00 (1H, dd, *J* = 3.2 and 10.8 Hz, H-3), 4.45 (1H, d, *J* = 3.7 Hz, H-4), 4.31 (1H, dd, *J* = 11.1 and 6.8 Hz, H-6a), 4.21 (1H, dd, *J* = 11.2 and 6.8 Hz, H-6b), 3.98 (1H, m, H-5), 3.45 (3H, s, 1-OC*H*_3_); LC-MS [M+1]^+^ 1052.29; Anal Calcd. for C_73_H_62_O_7_: C, 83.40%, H, 5.94%; found: C, 83.41%, H, 5.96%. 

Methyl 6-*O*-cinnamoyl-2,3,4-tri-*O*-(4-nitrobenzoyl)-β-D-galactopyranoside (9): IR (KBr): ν/cm^−1^ 1711 (C=O), 1635 (-CH=CH-); ^1^H-NMR (400 MHz, CDCl_3_) (ppm): δ_H_ 8.17, 7.62 {3 ×(2×2H), 3 × d, *J* =8.7 Hz_,_ Ar-H}, 7.65 (1H, d, *J* = 12.0 Hz, PhC*H*=CHCO-), 7.36 (2H, m, Ar-H), 7.31 (3H, m, Ar-H), 6.38 (1H, d, *J* = 12.1 Hz, PhCH=C*H*CO-), 4.96 (1H, d, *J* = 8.1 Hz, H-1), 4.88 (1H, dd, *J* = 8.1 and 10.6 Hz, H-2), 4.67 (1H, dd, *J* = 3.0 and 10.6 Hz, H-3), 4.47 (1H, d, *J* = 3.6 Hz, H-4), 4.28 (1H, dd, *J* = 11.1 and 6.6 Hz, H-6a), 4.00 (1H, dd, *J* = 11.0 and 6.7 Hz, H-6b), 3.96 (1H, m, H-5), 3.51 (3H, s, 1-OC*H*_3_); LC-MS [M+1]^+^ 772.64; Anal Calcd. for C_37_H_29_N_3_O_16_: C, 57.59, H, 3.78%; found: C, 57.61, H, 3.79%.

Methyl 2,3,4-tri-*O*-(4-bromobenzoyl)-6-*O*-cinnamoyl-β-D-galactopyranoside (10): IR (KBr): ν/cm^−1^ 1680 (C=O), 1635 (-CH=CH-); ^1^H-NMR (400 MHz, CDCl_3_) (ppm): δ_H_ 8.01 (3 ×2H, m, Ar-H), 7.93 (3 ×2H, m, Ar-H), 7.51 (1H, d, *J* = 12.0 Hz, PhC*H*=CHCO-), 7.31 (2H, m, Ar-H), 7.28 (3H, m, Ar-H), 6.62 (1H, d, *J* = 12.1 Hz, PhCH=C*H*CO-), 5.20 (1H, d, *J* = 8.2 Hz, H-1), 5.00 (1H, dd, *J* = 8.0 and 10.5 Hz, H-2), 4.81 (1H, dd, *J* = 3.1 and 10.6 Hz, H-3), 4.42 (1H, d, *J* = 3.7 Hz, H-4), 4.08 (1H, dd, *J* = 11.0 and 6.5 Hz, H-6a), 3.98 (1H, dd, *J* = 11.1 and 6.8 Hz, H-6b), 3.88 (1H, m, H-5), 3.56 (3H, s, 1-OC*H*_3_); LC-MS [M+1]^+^ 874.33; Anal Calcd. for C_37_H_29_O_10_Br_3_: C, 50.88, H, 3.34%; found: C, 50.87, H, 3.35%.

### 3.4. Biological Assessment

In this work, some acylated derivatives of methyl-β-D-galactopyranoside were selected and screened for the antibacterial activities against five human pathogenic bacteria (viz. *Bacillus subtilis* (ATCC 6633), *Staphylococcus aureus* (ATCC 6538), *Escherichia coli* (ATCC 8739), *Salmonella abony* (NCTC 6017), and *Pseudomonas aeruginosa* (ATCC 9027). The tested chemicals were also screened for antifungal activities against two human pathogenic fungi (viz. *Aspergillus niger* (ATCC 16404) and *Aspergillus flavus* (ATCC 10231). The test chemicals’ minimum inhibitory concentration (MIC) and minimum bactericidal concentration (MBC) were determined against these organisms**.** The potential activity of the methyl β-D-galactopyranoside (**1**) a against Ehrlich’s ascites carcinoma (EAC) cancer cell was also examined in this work.

#### 3.4.1. Collection of Bacterial Strains and Fungus

Test tube cultures of five bacterial and fungal pathogens were obtained from the Microbiology Laboratory, Department of Microbiology, University of Chittagong. The bacterial strains were allowed at 25 °C in nutrient agar medium (peptone 0.5%, beef extract 0.3%, NaCl 0.5%, agar 1.5%, and distilled water, and pH is adjusted to neutral 7.4).

#### 3.4.2. Antibacterial Activity 

The newly acylated methyl β-D-galactopyranoside (**1**) analogs in vitro were selected and screened for the antibacterial investigation against five human pathogenic bacteria by the disc diffusion method [41]. This method used paper discs of 4 mm diameter and a glass Petri-plate of 90 mm in diameter throughout the experiment. Sterile 5% (*w*/*v*) dimethyl sulfoxide (DMSO) solution was used to prepare a desired concentrated solution of the synthesized compounds and standard antibiotics. The paper discs were soaked with test chemicals of concentration 20 mg/mL for antibacterial analysis. The bacterial suspensions were swabbed with Mueller-Hinton agar media (MHA) and the sterile soaked discs were placed on it. The plates were incubated at 37 °C for the growth of test organisms and observed after 24 h. Azithromycin from BEXIMCO (Bangladesh) Ltd. (Dhaka, Bangladesh) was used as a positive control, while DMSO was negative. 

#### 3.4.3. Determination of MIC and MBC

A multiwell plate (96 wells) was used to determine the minimum inhibitory concentration (MIC) as the guidelines adopted by the Clinical and Laboratory Standards Institute (CLSI) [42]. Tests were carried out in Mueller-Hinton broth at pH 7.4, and the twofold serial dilution technique was applied. A standardized suspension of microorganisms was prepared for use with a 0.5 tube on the McFarland scale. The plate was incubated for 24 h, an indicator 10 µL 2, 3, 5-triphenyltetrazolium chloride 0.5% (*w*/*v*) solution was applied to show microbial growth in the well. The MIC was calculated as the concentration in the last well where no microbial growth occurred. The contents of the wells were sown on plates with Mueller-Hinton agar medium to determine the minimum bactericidal concentration (MBC), which is the concentration at which no colony growth occurs. All analyses were carried out in triplicate. The 1st well was treated as a negative control with no chemicals, while the 8th well was treated as a positive control with the standard antibiotic azithromycin. 

#### 3.4.4. Aantifungal Evaluation

The “Food poisoned” technique was employed for the determination of mycelial growth [43,44] of the synthesized methyl β-D-galactopyranoside (**1**) analogs against two fungi. First, the sterilized melted Potato Dextrose Agar medium (PDA at 45 °C) was poured at the rate of 20 mL in each sterilized petri dish (90 mm). After solidifying the medium, the fungal inoculums (5mm mycelial block) were placed in the center of the petri dishes, after 48 h. Then, incubation at 37 °C, the percentage inhibition of mycelial growth of the fungi was calculated as the following equation.
$$\;\text{I}\;=\frac{C-T}{C}\;\times 100$$
where, I = percentage of inhibition; C = diameter of the fungal colony in control (DMSO)
T = diameter of the fungal colony in treatment

Positive control was maintained with Nystatin, and negative control was also maintained without using any chemicals.

#### 3.4.5. Anticancer Activity 

Adult Swiss albino mice were collected from the International Center for Diarrhoeal Disease Research, Bangladesh. In vivo proliferation of Ehrlich’s ascites carcinoma (EAC) cells was performed according to Ahmed et al. [45]. The cells were collected from the mice, and their viability was checked by trypan blue exclusion assay. In addition, MTT colorimetric assay was used to detect the in vitro proliferation of EAC cells. Viable EAC cells (5 × 10^5^ in 100 μL RPMI-1640 media) were placed in a 96-well flat-bottom culture plate in the presence and absence of different concentrations of compound **9** (4-nitrobenzoyl derivative of methyl β-D-galactopyranoside) (62.5–500 μg/mL) and incubated at 37 °C in a CO_2_ incubator for 24 h. After removing the aliquot from each well, 10 mM of PBS (180 μL) and MTT (20 μL, 5 mg/mL MTT in PBS) were added and incubated at 37 °C for 4 h. Then the aliquot was removed again, and 200 μL of acidic isopropanol was added to each well. The plate was agitated for 5 min and incubated at 37 °C for 1 h, and finally, the absorbance was taken at 570 nm using a titer plate reader. The following equation was applied to calculate the cell proliferation inhibition ratio: Proliferation inhibition ratio (%) = {(A − B) × 100}/A
where A is the OD_570_ nm of the cellular homogenate (control) without compound (**9**) and B is the OD_570_ nm of the cellular homogenate with compound (**9**).

### 3.5. PASS Parameter Evaluation

The online web application PASS (http://www.pharmaexpert.ru/passonline/ (accessed on 11 July 2021) has been employed to calculate the antimicrobial activity spectrum of the selected MGP ester [46]. Firstly, the MGP analogs structures were drawn and then, changed into their smiles formats by using SwissADME free online applications (http://www.swissadme.ch (accessed on 11 July 2021), which were renowned to determine antimicrobial spectrum using the PASS web tool. PASS outcomes are revealed by Pa (probability for active molecule) and Pi (probability for inactive molecule). Having potentialities, the Pa and Pi scores vary in the range of 0.00 to 1.00 and usually, Pa + Pi ≠ 1, as these potentialities are predicted freely. The biological actions with Pa > Pi are only thought of as probable for a selected drug molecule.

### 3.6. Computational Details

Quantum mechanical methods are widely used in computational chemistry to calculate thermal, molecular orbital, and molecular electrostatic properties [47]. The Gaussian 09 program [48] is used to optimize the geometry and modify all synthesized analogs further. Density functional theory (DFT) with Beck’s (B) [49] three-parameter hybrid model, Lee, Yang, and Parr’s (LYP) [50] correlation functional under 3–21 G basis set has been employed to optimize and predict their thermal and molecular orbital properties. Dipole moment, enthalpy, Gibb’s free energy and electronic energy were calculated for all the compounds. For each of the MGP analogs, HOMO-LUMO energy gap, hardness (*η*), and softness (*S*) were calculated from the energies of frontier HOMO and LUMO as reported considering Parr and Pearson interpretation of DFT and Koopmans theorem [51]. The following equations are used to calculate hardness (*η*), softness (*S*), and chemical potential (*µ*).
$$\mathrm{Gap}\;\left(\Delta_{\varepsilon }\right)=\varepsilon \mathrm{LUMO}-\varepsilon \mathrm{HOMO};$$
$$\eta =\frac{\left[\varepsilon \mathrm{LUMO}-\varepsilon \mathrm{HOMO}\right]}{2};$$
$$S=\frac{1}{\eta };$$
$$µ=\frac{\left[\varepsilon \mathrm{LUMO}+\varepsilon \mathrm{HOMO}\right]}{2}$$

### 3.7. Preparation of Protein and Molecular Docking

The 3D structure of SARS-CoV-2 M^pro^ (PDB ID. 6Y84) (Appendix A) was collected in pdb format from the protein data bank [52]. All hetero atoms and water molecules were taken away by using PyMol (version 1.3) software packages [53]. Swiss-Pdb viewer software (version 4.1.0) was employed for energy minimization of the protein [54]. In fine, molecular docking simulation was rendered by PyRx software (version 0.8) [55], considering the protein as a macromolecule and the drug as ligand. The grid box size in AutoDockVina was kept at 37.0771, 63.9808, and 62.9744 Å for X, Y, Z directions, respectively. Accelrys Discovery Studio (version 4.1) was used to explore and visualize the docking result and search the non-bonding interactions between ligands and amino acid residues of receptor protein [56].

PDBsum online server was also used to check the validation of the protein (PDB: 6Y84) with Ramachandran (Appendix A) and Lig-plot (Appendix A), which revealed that 89.60% residues in the allowed region and no residues were missed.

### 3.8. Molecular Dynamics Simulation

The AMBER14 force field was used to run the YASARA dynamics software package [57,58]. Initially, the docked complexes were cleaned and optimized, and hydrogen bond network systems were oriented. In a cubic simulation cell with periodic boundary conditions, the TIP3P solvation model was used [59]. The physiological parameters of the simulation systems were set as 310 K temperature, pH 7.4, and 0.9% NaCl. The initial energy minimizations of the systems were done by the steepest gradient approaches (5000 cycles) by simulated annealing methods [60]. The Particle Mesh Ewalds method calculated the long-range electrostatic interactions by a cut-off radius of 8.0Å [61,62,63]. The time step of the simulations systems was set as 2.0 fs. The simulation trajectories were saved after every 100 ps. By following constant pressure and Berendsen thermostat, the simulation was conducted for 150 ns. The simulation trajectories were utilized to calculate the root mean square deviations and root means square fluctuations, solvent accessible surface area, gyration radius, and hydrogen bond [64,65,66,67].

### 3.9. Pharmacokinetic Prediction

To study pharmacokinetic parameters and toxicity of the MGP analogs the admetSAR server was utilized. We have utilized the admetSAR online database to evaluate the pharmacokinetics parameters related to the parent drug’s drug absorption, metabolism, and toxicity and its designed analogs [68]. Using structure similarity search methods, admetSAR predicts the latest and most comprehensive manually curated data for diverse chemicals associated with known ADME/T profiles. Generally, drug-likeness is evaluated using Lipinski’s rule of five [69]. Moreover, physicochemical properties are studied by utilizing the SwissADME server.

## 4. Conclusions

In conclusion, the inherent characteristic stability and biochemical behavior of MGP and synthesized analogs were investigated. The most significant properties for biological chemistry, chemical reactivity, and frontier orbital study like PASS, HOMO, LUMO, gap, and molecular electrostatic potential in molecule were optimized, indicating that it may be a good drug molecule. Because all of the designed MGP analogs have a smaller HOMO-LUMO gap than MGP, the modified compounds are more reactive than the parent drug. Incorporating various aliphatic and aromatic groups into the structure of MGPs can significantly improve their biological activity mode. The study found that nitro and bromo substituted benzoyl analogs **9**–**10** of MGP had greater antifungal activity and better pharmacokinetics and biological spectra. These findings were supported by molecular docking, which revealed promising antiviral efficacy MGP analogs. Many of these analogs showed remarkable binding interactions and binding energy with SARS-CoV-2 M^pro^. The five MGP analogs (**3**–**4** and **7**–**10**) have the potent ability in silico to fight SARS-CoV-2. The molecular electrostatic potential study also showed the most negative and positive surface area of the investigated ligand and anticipated the suitable site for hydrogen bonding. This result extremely and decidedly supports conducting molecular dynamics studies for up to 150 ns, keeping a pocket of protein which confirms the binding stability of the docked complex in the trajectory analysis, meaning that the protein–ligand complex is highly stable in a biological system. In fine, these analogs were analyzed for their pharmacokinetic properties, which expressed that the combination of toxicity prediction*,* in silico ADMET prediction, and drug-likeness had promising results because most of the designed molecules have improved kinetic parameters. It maintains all drug-likeness rules as well as an interesting result in terms of biological activity. Finally, this research may be useful to understand the chemical, thermal, physicochemical, biological, and pharmacokinetic properties of MGP analogs. As this study has been carried out using synthetic, antimicrobial, anticancer and in silico computational methods, therefore being particular about these results would require further wet-lab experiments to be carried out under in vivo and in vitro conditions if these analogs could be drugs candidates to treat SARS-CoV-2.

## Data Availability

Data is available in this article and Appendix A.

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
