# Peer review of "Synthesis, Antimicrobial, Anticancer, PASS, Molecular Docking, Molecular Dynamic Simulations & Pharmacokinetic Predictions of Some Methyl β-D-Galactopyranoside Analogs"

_molecules, 2021, doi:10.3390/molecules26227016_

Round 1
Reviewer 1 Report
The manuscript - molecules-1421643-peer-review-v1 - “Synthesis, Antimicrobial, Anticancer, PASS, Molecular Docking, Molecular Dynamic Simulations and Pharmacokinetic Predictions of some Methyl β-D-Galactopyranoside Analogs” is interesting and the theme of the article meets the scope of the Molecules journal.
This is another paper in the long list appeared this year with the objectives of searching for active components against Covid-19 virus. The paper is definitely an important contribution to these efforts.
The manuscript can be very easily improved by addressing the following points:
- The abstract is too long. I strongly recommend rewriting the abstract using short sentences and highlighting the main purpose and the usefulness of the results. Please see Instructions for Authors “The abstract should be a total of about 200 words maximum.”
- Could you please simplify the manuscript and underline the most important parts regarding the objectives of Manuscript? The authors used through the manuscript text too many details to describe the software or methods. The information are very often repeated. In this context, I strongly suggest to present only the meaningful information regarding the methods and software's and to rewrite some sections in order to be easy to follow.
- In Tables 4-12 the word “Drug” is appropriate?
- Some materials can be presented as supplementary material (e.g. Figures 11, 12, 14, 16, 17, and 18, and some tables). The representations of hydrogen bond and hydrophobic interactions is would help the readers (e.g. in Fig.12).
- Regarding the prediction section of antimicrobial activity, of molecular descriptors, my suggestion is that the authors present all calculated descriptors with their acronyms or explanations as separate Tables in the Supplementary information. In the main text please present only the significant information/results.
- With modern computational resources, several hundred ns simulations could be performed within several days. I strongly recommend longer MD simulation runs to explore more accurately the ligand-binding mode.
- However, there is great potential for the work, as long as its structure is redone, the results and discussion are improved, adequate justifications are made, and a good conclusion paragraph that reinforces and makes clear the contribution of this research.
Author Response
Manuscript ID: molecules-1421643
List of corrections
We corrected according to the reviewer’s suggestions as our attachment files.

Reviewer 2 Report
In this review, the authors report the synthesis, diverse in silico evaluations and simple biological tests of new Galactopyranoside analogues.
The newly reported monosaccharides are characterised by the presence of an ester junction with a cinnamoyl group on position 6 and various acylating agents on positions 2 to 4 of the methyl-b-D-galactopyranoside. The products are tested at very high concentrations for antimicrobial, antifungal and anticancer activities using very simple and generic tests. Several complementary in silico studies are reported to foresee potential antiviral activity on SARS-CoV-2 using potential docking on 6Y84 protease as well as diverse molecular physicochemical and molecular dynamics data. An in silico ADMET study is also reported to complement the overall study.
The results reported by the authors do not show particular significant activity of the new analogues either on microbiology or cytotoxic experiments. The evaluation of the interaction on the viral protease might be interesting but should be complemented, at least, by an in vitro evaluation on an infected cellular model.
Many figures are redundant or unnecessary. For example, Figures 1, 3, 4, 5 and 8 are redundant with the Tables. Percentages of cell growth inhibition obtained with compound 9 remain quite low and might be reported in a single phrase.
IR and NMR results are well reported in the Material and Methods section, Figure 2 can then be deleted.
In the Supplementary section, S1 and S4 do not add specific value to the manuscript and might be deleted.
Author Response
Manuscript ID: molecules-1421643
List of corrections
We corrected according to the reviewer’s suggestions as attachment files.

Round 2
Reviewer 1 Report
The authors have satisfactorily responded to all my questions and made the necessary changes to the manuscript. I have no further comment on the revised version of the manuscript “Synthesis, Antimicrobial, Anticancer, PASS, Molecular Docking, Molecular Dynamic Simulations and Pharmacokinetic Predictions of some Methyl β-D-Galactopyranoside Analogs”. I believe the work is now in a publishable form.
Reviewer 2 Report
The authors propose a largely revised version of the manuscript. In this revised version most of all required modifications have been implemented, except some complementary in vitro tests to assess the possible interaction with protease 6Y84.